# `GLBench`: A Comprehensive Benchmark for Graph with Large Language Models

**Yuhan Li**[1*]**, Peisong Wang**[2*]**, Xiao Zhu**[1]**, Aochuan Chen**[1]**, Haiyun Jiang**[3†]**, Deng Cai**[4]**, Victor Wai Kin Chan**[2]**, Jia Li**[1†]

[1] Hong Kong University of Science and Technology (Guangzhou) [2] Tsinghua University
[3] School of Electronic Information and Electrical Engineering, Shanghai Jiao Tong University [4] Tencent AI Lab
✉ Primary contact: `jialee@ust.hk`

## Abstract

The emergence of large language models (LLMs) has revolutionized the way we interact with graphs, leading to a new paradigm called GraphLLM. Despite the rapid development of GraphLLM methods in recent years, the progress and understanding of this field remain unclear due to the lack of a benchmark with consistent experimental protocols. To bridge this gap, we introduce `GLBench`, the first comprehensive benchmark for evaluating GraphLLM methods in both supervised and zero-shot scenarios. `GLBench` provides a fair and thorough evaluation of different categories of GraphLLM methods, along with traditional baselines such as graph neural networks. Through extensive experiments on a collection of real-world datasets with consistent data processing and splitting strategies, we have uncovered several key findings. Firstly, GraphLLM methods outperform traditional baselines in supervised settings, with LLM-as-enhancers showing the most robust performance. However, using LLMs as predictors is less effective and often leads to uncontrollable output issues. We also notice that no clear scaling laws exist for current GraphLLM methods. In addition, both structures and semantics are crucial for effective zero-shot transfer, and our proposed simple baseline can even outperform several models tailored for zero-shot scenarios. The data and code of the benchmark can be found at https://github.com/NineAbyss/GLBench.

## 1 Introduction

Graphs are ubiquitous in modeling the relational and structural aspects of real-world objects, encompassing a wide range of real-world scenarios [58, 11, 60, 59, 50, 65]. Many of these graphs have nodes that are associated with text attributes, resulting in the emergence of text-attributed graphs, such as citation graphs [17, 36] and web links [37]. For example, in citation graphs, each node represents a paper, and its textual description (e.g., title and abstract) is treated as the node's text attributes.

To tackle graphs with both node attributes and graph structural information, conventional pipelines typically fall into two categories. Firstly, graph neural networks (GNNs) [24, 12, 27, 57, 33] have emerged as the dominant approach through recursive message passing and aggregation mechanisms among nodes. They often utilize non-contextualized shallow embeddings, such as bag-of-words [13] and skip-gram [38], as shown in the previous benchmarks [42, 17]. Secondly, pretrained language models (PLMs) can be directly employed to encode the text associated with each node, transforming the problem into a text classification task without considering graph structures [61, 8]. Following [52], we refer to these two learning paradigms as **GNN-based** and **PLM-based methods**.

In recent years, the advent of large language models (LLMs) with massive context-aware knowledge and semantic comprehension capabilities (e.g., LLaMA [46], GPTs [1], Mistral [20]) marks a

---

[*]Equal contribution
[†]Corresponding author

38th Conference on Neural Information Processing Systems (NeurIPS 2024) Track on Datasets and Benchmarks.

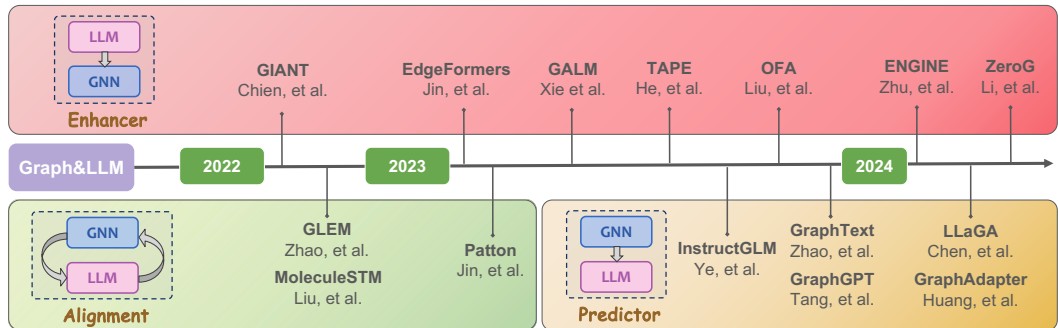

Figure 1: Timeline of GraphLLM research. Existing methods can be divided into three categories based on the role played by LLM. Top left corner illustrates the key differences of roles.

significant advancement in AI research, also leading a notable shift in the way we interact with graphs [7]. For example, LLMs can retrieve semantic knowledge that is relevant to the nodes to improve the quality of initial node embeddings of GNNs [15, 49]. LLMs can also encode graphs through carefully designed prompts and directly make predictions in an autoregressive manner [5, 18]. In addition, LLMs can be aligned with GNNs in the same vector space, enabling GNNs to be more semantically aware [61, 32]. In this paper, we refer to this new paradigm of using LLMs to assist with graph-related problems as **GraphLLM**. We follow existing surveys [25, 21] to organize existing GraphLLM methods into three categories based on the role (i.e., enhancer, predictor, and aligner) played by LLMs throughout the entire model pipeline.

Despite the plethora of GraphLLM methods proposed in recent years, as illustrated in Figure 1, there is no comprehensive benchmark for GraphLLM, which significantly impedes the understanding and progress of existing methods in several aspects. *i)* The use of different datasets, data processing approaches, and data splitting strategies in previous works makes many of the results incomparable, leading a lack of comparison and understanding of different categories of GraphLLM methods, i.e., which roles LLMs are better suited to play. *(ii)* The lack of benchmarks for zero-shot graph learning has led to limited exploration in this area. *(iii)* Apart from effectiveness, understanding each method's computation and memory costs is imperative, yet often overlooked in the literature. A comparison of existing benchmarks with `GLBench` in terms of datasets, models, and scenarios is shown in Table 1.

To redress these gaps and foster academia-industry synergy in evaluation, we propose `GLBench`, which serves as the first comprehensive benchmark for this community in both supervised and zero-shot scenarios. Besides traditional GNN-based and PLM-based methods, we implement a wide range of existing GraphLLM models, encompassing LLM-as-enhancer models, LLM-as-predictor models, and LLM-as-aligner models. We adopt consistent data pre-processing and data splitting approaches for fair comparisons. Through extensive experiments, our key insights include *(i)* GraphLLM methods exhibit superior performance across the majority of datasets, with LLM-as-enhancers demonstrating the most robust performance; *(ii)* The performance of LLM-as-predictor methods is not as satisfactory, and they often encounter uncontrollable output issues; *(iii)* There is no obvious scaling law in existing methods; *(iv)* Structural and semantic information are both important for zero-shot transfer; *(v)* We propose a training-free baseline, which can even outperform existing GraphLLM methods tailored for zero-shot scenarios. In summary, we make the following three contributions:

- We introduce `GLBench`, the first comprehensive benchmark for GraphLLM methods in both supervised and zero-shot scenarios, enabling a fair comparison among different categories of methods by unifying the experimental settings across a collection of real-world datasets.

- We conduct a systematic analysis of existing methods from various dimensions, encompassing supervised performance, zero-shot transferability, and time and memory efficiency.

- Our benchmark repository is publicly available at https://github.com/NineAbyss/GLBench to facilitate future research on GraphLLM and graph foundation models.

## 2 Formulations and Background

In this section, we introduce the foundational concepts related to `GLBench`, including task definitions, learning scenarios, and the advances of GraphLLM.

Table 1: Comparison of existing Graph benchmarks in terms of datasets, models, and scenarios. We focus on constructing a benchmark for GraphLLM methods, encompassing different categories of GraphLLM methods as well as traditional GNN- and PLM-based methods. In addition, we explore the zero-shot scenario, which is often overlooked by previous benchmarks.

| Benchmark | #Datasets (Node-level) | #Domains | Text | #Models (GraphLLM) | Model Type | Supervision Scenario |
|---|---|---|---|---|---|---|
| Sen et al. [42] | 2 (2) | 1 | ✗ | 8 (0) | Classical | Supervised |
| Shchur et al. [43] | 8 (8) | 2 | ✗ | 8 (0) | GNN | Supervised |
| OGB [17] | 14 (5) | 3 | ✗ | 20 (0) | GNN | Supervised |
| CS-TAG [52] | 8 (6) | 2 | ✓ | 16 (2) | GNN, PLM, Enhancer | Supervised |
| GLBench | 7 (7) | 3 | ✓ | 18 (12) | GNN, PLM, GraphLLM | Supervised and Zero-shot |

## 2.1 Notations

In this benchmark, we focus on the node classification task within the text-attributed graphs (TAGs). Formally, a TAG can be represented as $\mathcal{G} = (\mathcal{V}, A, \{s_n\}_{n \in \mathcal{V}})$, where $\mathcal{V}$ is a set of $N$ nodes; $A \in \{0, 1\}^{N \times N}$ is the adjacency matrix, if $v_i$ and $v_j$ are connected, $A_{ij} = 1$, otherwise $A_{ij} = 0$; $s_n \in \mathcal{D}^{L_n}$ is a textual description associated with node $n \in \mathcal{V}$, with $\mathcal{D}$ as the words or tokens dictionary and $L_n$ as the sequence length. Typically, in most previous graph machine learning literature, such attribute information can be encoded into shallow embeddings $\mathbf{X} = [\mathbf{x}_1, \mathbf{x}_2, \ldots, \mathbf{x}_N] \in \mathbb{R}^{N \times f}$ by naive methods (*e.g.*, bag-of-words or TF-IDF [41]), where $f$ is the dimension of embeddings. Given TAGs with a set of labeled nodes $\mathcal{L}$ (where $\mathcal{L}$ can be empty), the goal of node classification is to predict the labels of the remaining unlabeled nodes.

## 2.2 Learning Scenarios

GLBench supports two learning scenarios. The first scenario is **supervised learning**, which aims to train GraphLLM models to predict the unlabeled nodes with the same label space of training set on a single graph. Formally, let $\mathcal{G}_s = (\mathcal{V}_s, A_s, \mathbf{X}_s)$ with label space $\mathcal{Y}_s$ be the source graph used for training, and let $\mathcal{G}_t = (\mathcal{V}_t, A_t, \mathbf{X}_t)$ with label space $\mathcal{Y}_t$ be the target graph used for testing. The supervised learning scenario can be denoted as $\mathcal{G}_s = \mathcal{G}_t, \mathcal{Y}_s = \mathcal{Y}_t$. It is noted that supervised learning can be further divided into fully- and semi-supervised settings based on the ratio of training samples. In this paper, we do not make this finer distinction and instead refer to both as supervised learning. The detailed training ratios of each dataset can be found in Table 3.

The second scenario is **zero-shot learning**, which is an emerging topic in the field of graph machine learning. The goal of this setting is to train GraphLLM models on labeled source graphs and generate satisfactory predictions on a completely different target graph with distinct label spaces, which can be denoted as $\mathcal{G}_s \cap \mathcal{G}_t = \emptyset, \mathcal{Y}_s \cap \mathcal{Y}_t = \emptyset$. According to existing works [29, 26], zero-shot learning poses inherent challenges such as feature misalignment and mismatched label spaces compared with traditional supervised learning. For example, different datasets often use varying shallow embedding techniques, leading to incompatible feature dimensions across datasets and making it difficult for a model trained on one dataset to adapt to another. We are the first to establish a benchmark for such a practical scenario. While only a few GraphLLM methods are capable of performing zero-shot inference, we believe this setting is crucial and well-suited for the graph foundation model era, emphasizing broad generalization across different graph sources [35, 30].

## 2.3 LLMs for Graphs

To provide a global understanding of the literature, we follow [25, 21] to categorize the methods we evaluated in our benchmark based on the **roles** played by LLMs throughout the entire model pipeline. Table 2 provides an overview of the models assessed in GLBench. Specifically, LLM-as-enhancer approaches correspond to enhancing the quality of node embeddings or textual attributes with the help of LLMs. For example, TAPE [15] leverages LLMs to enrich initial node embedding in GNNs with semantic knowledge relevant to the nodes. In addition, several methods, such as GraphGPT [44] and LLaGA [6], utilize LLMs as predictors to perform classification within a unified generative paradigm. Lastly, GNN-LLM alignment ensures that each encoder's unique capabilities are preserved while coordinating their embedding spaces at a specific stage. For instance, GLEM [61] employs an

Table 2: A summary of all GraphLLM methods used in our evaluation, which can be divided into three categories based on the roles (i.e., enhancer, predictor, and aligner) played by LLMs. **Fine-tune** denotes whether it is necessary to fine-tune the parameters of LLMs, and **Prompt** indicates the use of text-formatted prompts in LLMs, done manually or automatically.

| Role | Method | Predictor | GNN | PLM/LLM | Techniques Used | | Learning Scenarios | | Venue | Code |
|------|--------|-----------|-----|---------|-----------------|--------|--------------------|----------|-------|------|
| | | | | | Fine-tune | Prompt | Supervised | Zero-shot | | |
| Enhancer | GIANT [10] | GNN | GraphSAGE, etc. | BERT | ✗ | ✗ | ✓ | ✗ | ICLR'22 | Link |
| | TAPE [15] | GNN | RevGAT | ChatGPT | ✗ | ✓ | ✓ | ✗ | ICLR'24 | Link |
| | OFA [29] | GNN | R-GCN | Sentence-BERT | ✗ | ✓ | ✓ | ✓ | ICLR'24 | Link |
| | ENGINE [64] | GNN | GraphSAGE | LLaMA-2 | ✓ | ✓ | ✓ | ✗ | IJCAI'24 | Link |
| | ZeroG [26] | GNN | SGC | Sentence-BERT | ✓ | ✓ | ✗ | ✓ | KDD'24 | Link |
| Predictor | InstructGLM [55] | LLM | - | FLAN-T5/LLaMA-v1 | ✓ | ✓ | ✓ | ✗ | EACL'24 | Link |
| | GraphText [62] | LLM | - | ChatGPT/GPT-4 | ✓ | ✓ | ✓ | ✗ | Arxiv | Link |
| | GraphAdapter [19] | LLM | GraphSAGE | LLaMA-2 | ✓ | ✓ | ✓ | ✗ | WWW'24 | Link |
| | GraphGPT [44] | LLM | GT | Vicuna | ✓ | ✓ | ✓ | ✓ | SIGIR'24 | Link |
| | LLaGA [6] | LLM | - | Vicuna/LLaMA-2 | ✓ | ✓ | ✓ | ✗ | ICML'24 | Link |
| Aligner | GLEM [61] | GNN/LLM | GraphSAGE, etc. | RoBERTa | ✓ | ✗ | ✓ | ✗ | ICLR'23 | Link |
| | Patton [22] | LLM | GT | BERT/SciBERT | ✓ | ✗ | ✓ | ✗ | ACL'23 | Link |

EM framework to merge GNNs and LLMs, where one model iteratively generates pseudo-labels for the other. More detailed descriptions of models are provided in Appendix A.2.

# 3 The Setup of `GLBench`

In this section, we begin by introducing the datasets utilized in `GLBench`, along with the method implementations. Then, we outline the research questions that guide our benchmarking study.

## 3.1 Datasets and Implementations

**Dataset selection.** To provide a comprehensive evaluation of existing GraphLLM methods, we gather 7 diverse and representative datasets, as detailed in Table 3, which are chosen based on the following criteria. *(i)* **Text-attributed graphs.** We only consider text-attributed graphs where each node represents a textual entity and is associated with a textual description. *(ii)* **Various domains.** Datasets in `GLBench` span multiple domains, including citation networks, web links, and social networks. *(iii)* **Diverse scale and density.** `GLBench` datasets cover a wide range of scales, from thousands to hundreds of thousands of nodes. The density also varies significantly, with average node degrees ranging from 2.6 (i.e., Citeseer) to 36.9 (i.e., WikiCS). Such diversity in datasets supports a robust evaluation of the effectiveness of GraphLLM methods in both supervised and zero-shot scenarios. Among the datasets in `GLBench`, Cora, Citeseer, Pubmed, and Ogbn-arxiv are designed to classify scientific papers into different topics. WikiCS aims to identify entities into different Wikipedia categories. Additionally, we incorporate two social networks that have been recently pre-processed by Huang et al. [19], i.e., Reddit and Instagram, which concentrate on identifying whether the user is a normal user. More descriptions of each dataset are shown in Appendix A.1.

**Dataset splits.** The data splitting methods in different works are not consistent, bringing difficulties in conducting fair comparisons. For example, a majority of GraphLLM methods [62] follow Yang et al. [54] to conduct experiments on the Cora dataset with the fixed 20 labeled training nodes per class at a 0.3% label ratio. However, several works [15, 55, 44] employ a 60%/20%/20% train/validation/test split, making it challenging to directly compare the performance with other methods. We investigate various GraphLLM works and choose the data splits that are most commonly used. More specifically, for Cora, Citeseer, and Pubmed, we use the classic split from [54, 24]. For Ogbn-arxiv, we follow the public partition provided by OGB [17]. For WikiCS, we follow the split in [37]. For two datasets from Huang et al. [19], we follow the original split described in the paper.

**Metrics.** According to existing node classification benchmarks [52, 17], we adopt *Accuracy* and *Macro-F1* score as the metrics for all datasets. Accuracy measures the overall correctness of the model's predictions, treating all classes equally. On the other hand, Macro-F1 score calculates the F1 score for each class independently and takes the average across all classes, making it more suitable for evaluating imbalanced datasets.

Table 3: Statistics of all datasets in `GLBench`, including the number of nodes and edges, the average node degree, the average number of tokens in the textual descriptions associated with nodes, the number of classes, the training ratio in the supervised setting, the node text, and the domains they belong to. More details are shown in Appendix A.1.

| Dataset | # Nodes | # Edges | Avg. # Deg | Avg. # Tok | # Classes | # Train | Node Text | Domain |
|---|---|---|---|---|---|---|---|---|
| **Cora** | 2,708 | 5,429 | 4.01 | 186.53 | 7 | 5.17% | Paper content | Citation |
| **Citeseer** | 3,186 | 4,277 | 2.68 | 213.16 | 6 | 3.77% | Paper content | Citation |
| **Pubmed** | 19,717 | 44,338 | 4.50 | 468.56 | 3 | 0.30% | Paper content | Citation |
| **Ogbn-arxiv** | 169,343 | 1,166,243 | 13.77 | 243.19 | 40 | 53.70% | Paper content | Citation |
| **WikiCS** | 11,701 | 216,123 | 36.94 | 642.04 | 10 | 4.96% | Entity description | Web link |
| **Reddit** | 33,434 | 198,448 | 11.87 | 203.84 | 2 | 10.00% | User's post | Social |
| **Instagram** | 11,339 | 144,010 | 25.40 | 59.25 | 2 | 10.00% | User's profile | Social |

**Implementations.** We rigorously reproduce all methods according to their papers and source codes. To ensure a fair evaluation, we perform hyperparameter tuning with a reasonable search budget on all datasets for GraphLLM methods. However, due to the considerably long training time of several LLM-as-predictor methods, i.e., InstructGLM [55] takes nearly 20 hours to train one epoch on the Arxiv dataset, we use the default parameters recommended in the paper for training. More details on these algorithms and implementations can be found in Appendix B.1.

## 3.2 Research Questions

### RQ1: Can GraphLLM methods outperform GNN- and PLM-based methods?

**Motivation.** Previous research in this community has been hindered by the use of different data preprocessing and splits, making it difficult to fairly evaluate and compare the performance of various GraphLLM methods with traditional methods. Given the fair comparison environment provided by `GLBench`, the first research question is to revisit the progress made by existing GraphLLM methods compared to using only GNNs or PLMs. By answering this question, we aim to gain a deeper understanding of the actual performance of existing GraphLLM methods, thereby providing insights into whether it is necessary to use GraphLLM methods as replacements for these traditional approaches in terms of effectiveness. In addition, as LLMs play different roles in GraphLLM methods, such as enhancer, predictor, and aligner, we are curious to investigate which roles are most effective in leveraging the power of LLMs for graph tasks.

**Experiment Design.** Following the experimental setting of most existing GraphLLM methods, we conduct the supervised node classification experiments on all the datasets in `GLBench`. For *(i)* **GNN-based methods**, we consider three prominent GNNs including GCN [24], GAT [47], and GraphSAGE [12]. For *(ii)* **PLM-based methods**, we select encoder-only PLMs with different sizes of parameters, including Sentence-BERT (22M) [40], BERT (110M) [23], and RoBERTa (355M) [34]. We avoid adopting decoder-only models for sentence encoding since they are specifically tuned for next-token prediction and do not perform as well in this scenario [26]. In addition, for a fair comparison and to minimize the influence of the backbone models used in different GraphLLM methods, we try to utilize the same GNN and LLM backbones in our implementations. For example, we consistently use GraphSAGE as GNNs for LLM-as-enhancer methods, and LLaMA2 as LLMs for any 7B-scale methods. More details can be found in Appendix B.2.

### RQ2: How much progress has been made in zero-shot graph learning?

**Motivation.** The goal of zero-shot graph learning, as discussed in Section 2.2, is to develop foundation models capable of generalizing to unseen graphs, thereby enabling transferability across diverse graph sources [9]. Traditional GNNs face challenges in such scenarios primarily due to dimension misalignment, mismatched label spaces, and negative transfer [26]. Recent efforts have been made to use LLMs directly as classifiers for zero-shot inference in graphs [8, 18]. For example, for each node (i.e., one paper) in a citation network, we can directly input the title and abstract of the paper into an LLM without any fine-tuning and then ask which category the paper belongs to. In addition, the emergence of incorporating LLMs with GNNs also offers promising solutions to these challenges, leveraging the unified graph representation learning and powerful knowledge transfer capabilities of

LLMs [29, 26, 63]. These developments motivate us to systematically evaluate models with zero-shot capabilities, providing insights into the progress made in this challenging and practical scenario.

**Experiment Design.** To answer this research question, we evaluate three GraphLLM methods with zero-shot capabilities on all datasets in `GLBench` for node classification, which involves an optional pre-training phase using *source datasets* that have non-overlapping classes with the *target datasets*. To ensure data distribution practices in NLP and CV, where source datasets are typically larger than target datasets, we select the two largest graphs in `GLBench`, i.e., Ogbn-arxiv and Reddit for pre-training, and use the remaining datasets for inference. We involve additional models as baselines including *(i)* **graph self-supervised learning** (SSL), which focus on the transferability of learned structures [28], such as DGI [48] and GraphMAE [16]; *(ii)* **LLMs**, which only consider semantic information for classification, such as LLaMA3-70B [46], GPT-3.5-turbo [39], GPT-4o [1], and DeepSeek-chat [3]. We follow [8] to design input prompts for each dataset; *(iii)* **Emb w/ NA**, a simple training-free baseline that utilizes both structural and semantic information for zero-shot inference.

**RQ3: Are existing GraphLLM methods efficient in terms of time and space?**

**Motivation.** As GraphLLM methods typically introduce more LLM parameters, even simultaneously optimizing GNNs and LLMs (e.g., as aligners), they inherently require more computational complexity and space than GNNs. However, the efficiency of GraphLLM methods has been largely overlooked in existing research. While introducing LLMs may benefit graph tasks, the extra computational consumption has posed significant requirements of trade-off between performance and efficiency. It is essential to understand the trade-off for deploying GraphLLM methods in practical applications.

**Experiment Design.** To answer this research question, we select two models from each of the three categories of GraphLLM methods and evaluate their efficiency in terms of time and space. Specifically, we record the wall clock running time and peak GPU memory consumption during each method's training process. We discuss efficiency based on datasets from multiple domains in `GLBench`, reporting results on Cora, Citeseer, WikiCS, and Instagram. For a fair comparison, all experiments in this part were conducted on a single NVIDIA A800 GPU.

## 4 Experiments and Analysis

### 4.1 Supervised Performance Comparison (RQ1)

Table 4 and Figure 2 shows the experimental results of various GraphLLM methods along with traditional GNN- and PLM-based methods under the supervised scenario. Our key findings include:

① **GraphLLM methods exhibit superior performance across the majority of datasets, highlighting their effectiveness.** From Table 4, we can observe that GraphLLM methods achieve the highest accuracy and F1 scores on 6 out of the 7 datasets compared to traditional GNN-based and PLM-based methods, particularly significant on several datasets such as Citeseer, Pubmed, and Reddit. For instance, the LLM-as-enhancer method OFA [29] outperforms the best performing GNN-based model GCN by 3.20% and 3.49% in terms of accuracy and F1 score respectively on Citeseer. In addition, the LLM-as-aligner approach PATTON [22] surpasses GCN by 5.18% in accuracy and 4.03% in F1 score on Pubmed. On the Reddit dataset, the LLM-as-predictor model LLaGA [6] achieves a 4.00% and 4.69% higher accuracy and F1 score compared to GCN. Notably, on the two largest graphs (i.e., Arxiv and Reddit), GraphLLM methods show stability with nearly all methods consistently outperforming baselines. The superior performance of GraphLLM methods can be attributed to their ability to jointly leverage both structure and semantics, enabling more powerful graph learning compared to traditional methods that only consider either structures or semantics independently. This highlights the necessity and benefit of incorporating LLMs in supervised learning.

② **The performance of LLM-as-predictor methods is not as satisfactory, and they often encounter uncontrollable output issues.** As evident from the notably bluer box plots in Figure 2, LLM-as-predictor methods generally underperform compared to traditional GNNs and other categories of GraphLLM methods across datasets. This performance drop is particularly evident in graphs with limited training data, such as Cora, Citeseer, and Pubmed, due to insufficient LLM training. However, LLM-as-predictor methods showcase better results on social network datasets. For instance, LLaGA [6] achieves the best results on Reddit, while GraphAdapter [19] achieves the second-best performance on Instagram. This can be attributed to the general and simpler semantics of textual attributes, such as user posts and profiles, which are easier for LLMs to comprehend. Furthermore,

Table 4: Accuracy and Macro-F1 results (%) under the supervised setting of `GLBench`. The highest results are highlighted with **bold** , while the second-best results are marked with underline .

| Model | Cora | | Citeseer | | Pubmed | | Ogbn-arxiv | | WikiCS | | Reddit | | Instagram | |
|---|---|---|---|---|---|---|---|---|---|---|---|---|---|---|
| | Acc | F1 | Acc | F1 | Acc | F1 | Acc | F1 | Acc | F1 | Acc | F1 | Acc | F1 |
| GCN [24] | **82.11** | **80.65** | 69.84 | 65.49 | 79.10 | 79.19 | 72.24 | 51.22 | 80.35 | 77.63 | 63.19 | 62.49 | 65.75 | 58.75 |
| GAT [47] | 80.31 | 79.00 | 68.78 | 62.37 | 76.93 | 76.75 | 71.85 | 52.38 | 79.73 | 77.40 | 61.97 | 61.78 | 65.38 | 58.60 |
| GraphSAGE [12] | 79.88 | 79.35 | 68.23 | 63.10 | 76.79 | 76.91 | 71.88 | 52.14 | 79.87 | 77.05 | 58.51 | 58.41 | 65.12 | 55.85 |
| Sent-BERT (22M) [40] | 69.73 | 67.59 | 68.39 | 64.97 | 65.93 | 67.33 | 72.82 | 53.43 | 77.07 | 75.11 | 57.31 | 57.09 | 63.07 | 56.68 |
| BERT (110M) [23] | 69.71 | 67.53 | 67.77 | 64.10 | 63.69 | 64.93 | 72.29 | 53.30 | 78.55 | 75.74 | 58.41 | 58.33 | 63.75 | 57.30 |
| RoBERTa (355M) [34] | 69.68 | 67.33 | 68.19 | 64.90 | 71.25 | 72.19 | 72.94 | 52.70 | 78.67 | 76.16 | 57.17 | 57.10 | 63.57 | 56.87 |
| GIANT [10] | 81.04 | 80.13 | 65.82 | 62.31 | 76.89 | 76.05 | 72.04 | 50.81 | 80.48 | 78.67 | 64.67 | 64.64 | 66.01 | 56.11 |
| TAPE [15] | 80.95 | 79.79 | 66.06 | 61.84 | 79.87 | 79.30 | 72.99 | 51.43 | 82.33 | 80.49 | 60.73 | 60.50 | 65.85 | 50.49 |
| OFA [29] | 75.24 | 74.20 | **73.04** | **68.98** | 75.61 | 75.60 | 73.23 | 57.38 | 77.34 | 74.97 | 64.86 | 64.95 | 60.85 | 55.44 |
| ENGINE [64] | 81.54 | 79.82 | 72.15 | 67.65 | 74.74 | 75.21 | 75.01 | 57.55 | 81.19 | 79.08 | 63.20 | 59.34 | **67.62** | **59.22** |
| InstructGLM [55] | 69.10 | 65.74 | 51.87 | 50.65 | 71.26 | 71.81 | 39.09 | 24.65 | 45.73 | 42.70 | 55.78 | 53.24 | 57.94 | 54.87 |
| GraphText [62] | 76.21 | 74.51 | 59.43 | 56.43 | 74.64 | 75.11 | 49.47 | 24.76 | 67.35 | 64.55 | 61.86 | 61.46 | 62.64 | 54.00 |
| GraphAdapter [19] | 72.85 | 70.66 | 69.57 | 66.21 | 72.75 | 73.19 | 74.45 | 56.04 | 70.85 | 66.49 | 61.21 | 61.13 | 67.40 | 58.40 |
| LLaGA [6] | 74.42 | 72.50 | 55.73 | 54.83 | 52.46 | 68.82 | 72.78 | 53.86 | 73.88 | 70.90 | **67.19** | **67.18** | 62.94 | 54.62 |
| GLEM_GNN [61] | **82.11** | 80.00 | 71.16 | 67.62 | 81.72 | 81.48 | **76.43** | **58.07** | **82.40** | **80.54** | 59.60 | 59.41 | 66.10 | 54.92 |
| GLEM_LLM [61] | 73.79 | 72.00 | 68.78 | 65.32 | 79.18 | 79.25 | 74.03 | 58.01 | 80.23 | 78.30 | 57.97 | 57.56 | 65.00 | 54.50 |
| PATTON [22] | 70.50 | 67.97 | 63.60 | 61.12 | **84.28** | **83.22** | 70.74 | 49.69 | 80.81 | 77.72 | 59.43 | 57.85 | 64.27 | 57.48 |

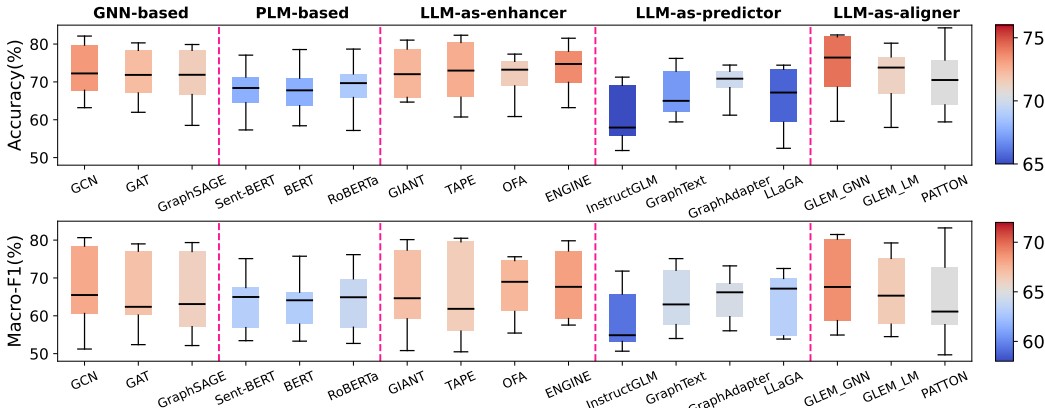

Figure 2: Comparison of the supervised performance among all methods. The color of the box plot represents the average score for each metric, while the central line within the box indicates the median score. We exclude results falling below 50% as they significantly deviate from the data center.

we observe that LLM-as-predictor methods often encounter uncontrollable output formats, despite employing instruction tuning. To maintain a lightweight tuning, existing methods typically freeze the parameters of LLMs and only fine-tune an adapter or projection layer. Such limited parameter space leads to uncontrollable and unexpected outputs (i.e., output incorrect or invalid labels), making it challenging to reliably extract the predicted labels and hinder the overall performance.

③ **Among the three categories, LLM-as-enhancer methods demonstrate the most robust performance across different datasets.** As shown in Figure 2, they achieve robust performance across datasets, considering both accuracy and F1 score. Among these methods, ENGINE [64] demonstrates the most outstanding effectiveness, consistently outperforming other methods by achieving either the best or the second-best results on 4 datasets. This highlights the benefits of employing LLMs to enrich the textual attributes or enhance the quality of node embeddings in supervised scenarios.

④ **Aligning GNNs and LLMs can improve the effectiveness of both. Specifically, the aligned GNNs are more reliable.** The alignment of GNNs and LLMs offers an effective method for integrating LLMs into graph learning. As illustrated in Table 4, aligned models, whether using GNNs or LLMs as classifiers, consistently outperform traditional GNN-based and PLM-based methods. This highlights the benefits of coordinating their embedding spaces at specific stages, which can

Table 5: Accuracy and Micro-F1 results (%) under the zero-shot setting of `GLBench`. The highest results are highlighted with **bold**, while the second highest results are marked with underline. $\mathcal{A}$ and $\mathcal{S}$ indicate whether structural and semantic information are used, respectively.

| Category | Model | $\mathcal{A}$ | $\mathcal{S}$ | Cora | | Citeseer | | Pubmed | | WikiCS | | Instagram | |
|---|---|---|---|---|---|---|---|---|---|---|---|---|---|
| | | | | Acc | F1 | Acc | F1 | Acc | F1 | Acc | F1 | Acc | F1 |
| Graph SSL | DGI [48] | ✓ | ✗ | 17.50 | 12.44 | 21.67 | 13.53 | 44.88 | 38.72 | 9.03 | 6.13 | **63.64** | 50.13 |
| | GraphMAE [16] | ✓ | ✗ | 27.08 | 23.66 | 15.24 | 14.44 | 22.03 | 15.65 | 10.74 | 6.69 | 53.56 | 52.18 |
| LLMs | LLaMA3 (70B) [46] | ✗ | ✓ | 67.99 | 68.05 | 51.44 | 49.98 | 77.00 | 64.18 | **73.64** | **72.62** | 38.23 | 36.41 |
| | GPT-3.5-turbo [39] | ✗ | ✓ | 65.67 | 63.22 | 50.58 | 49.34 | 75.99 | 69.90 | 68.75 | 66.56 | 49.39 | 49.67 |
| | GPT-4o [1] | ✗ | ✓ | **68.62** | **68.49** | 53.55 | 52.42 | 77.96 | 71.79 | 71.52 | 70.06 | 42.02 | 40.96 |
| | DeepSeek-chat [3] | ✗ | ✓ | 65.62 | 65.77 | 50.35 | 48.32 | **79.23** | 74.30 | 70.77 | 69.91 | 40.58 | 39.27 |
| Training-free | **Emb w/ NA** | ✓ | ✓ | 63.59 | 58.23 | 51.75 | 49.51 | 74.66 | 73.15 | 52.30 | 48.40 | 45.52 | 45.14 |
| Enhancer | OFA [29] | ✓ | ✓ | 23.11 | 23.30 | 32.45 | 28.67 | 46.60 | 35.04 | 34.27 | 33.72 | 53.63 | 51.10 |
| | ZEROG [26] | ✓ | ✓ | 62.52 | 57.53 | **58.92** | **54.58** | 79.08 | **77.94** | 60.46 | 57.24 | 56.13 | **52.50** |
| Predictor | GraphGPT [44] | ✓ | ✓ | 24.90 | 7.98 | 13.95 | 13.89 | 39.85 | 20.07 | 38.02 | 29.46 | 43.94 | 43.49 |

mutually enhance both performances. Notably, GLEM$_{\text{GNN}}$ [61] surpasses both GLEM$_{\text{LLM}}$ [61] and PATTON [22] across 6 of the 7 datasets evaluated, demonstrating that aligned GNNs are more reliable and better suited for supervised scenarios compared to aligned LLMs.

⑤ **There is no obvious scaling law in existing GraphLLM methods.** Recently, neural scaling law on graphs has become a focal point of discussion [4, 31], particularly in terms of two perspectives: data scaling and model scaling. Since we ensure consistency in the data, we only focus on the model scaling law. However, from Table 4, it can be observed that the performance does not clearly scale with model size. For instance, within the LLM-as-enhancer style, ENGINE [64] employs 7B-scale LLMs, yet it does not demonstrate a significant performance gap compared to the use of 110M-scale LLMs like OFA [29]. It is more noticeable in LLM-as-predictor methods, with models like InstructGLM [55] and GraphText [62] using 7B-scale LLMs yield unsatisfactory results, indicating that merely accumulating parameters does not guarantee better performance. In addition, we design a more rigorous and accurate experiment to better analyze the scaling law, as detailed in Appendix D. This experiment is based on different scales of GNNs and LLMs. The results indicate that it is challenging to identify clear scaling laws, whether related to the scaling of GNNs or LLMs, which is consistent with the insights presented here.

## 4.2 Zero-shot Transferability (RQ2)

We show the performance of GraphLLM methods with zero-shot capabilities, as well as baselines such as Graph SSL and LLMs in Table 5. We have the following observation based on the results:

⑥ **LLMs demonstrate strong zero-shot capabilities, but it may be due to data leakage.** Although completely disregarding structural information, LLMs still exhibit remarkable zero-shot capabilities, achieving the best results on 3 datasets. However, given that LLMs undergo pre-training on extensive text corpora, LLMs have likely seen and memorized at least part of the test data, especially for citation networks and wiki links [2, 51]. An interesting issue arises with Instagram, where LLMs often struggle to determine whether a user is normal and request more information related to the user, leading to errors. All logs of LLM zero-shot inference are released in our repository.

⑦ **Structural and semantic information are both important for zero-shot transfer.** Compared to graph SSL methods that only consider structure transfer, OFA [29] and ZeroG [26], which take into account both semantics and structure, achieve significant performance gains on all datasets. Transferring both structural and semantic information from the source datasets is crucial, ensuring that target datasets are not only enriched with the foundational structural patterns but also imbued with the contextual semantic nuances from the source datasets.

⑧ **A simple baseline can even outperform several existing GraphLLM methods tailored for zero-shot scenarios.** We propose a training-free baseline **Emb w/ NA** consisting of the following steps. Firstly, a frozen LLM is employed as a unified encoder to embed both node texts and class descriptions. Then, neighborhood aggregation is performed iteratively, updating node embeddings

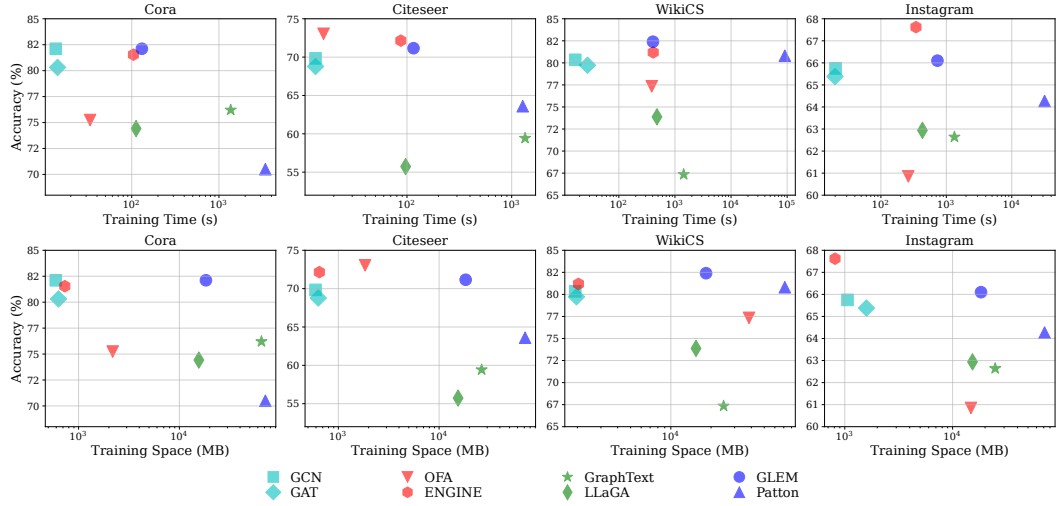

Figure 3: Training time and space analysis on Cora, Citeseer, WikiCS, and Instagram.

by aggregating features from their immediate neighbors. Finally, the class that yields the highest similarity score with the node embedding is predicted to be the class of the node. Through this simple design, we introduce both structural and semantic information at minimal cost. Surprisingly, as shown in Table 5, this training-free baseline can even outperform several GraphLLM methods tailored for zero-shot scenarios that require pre-training, such as OFA [29] and GraphGPT [44], across all datasets. The effectiveness of this baseline is further demonstrated by the fact that even a relatively small LLM (i.e., we use Sentence-Bert in Table 5) can achieve impressive results. This finding motivates the potential of developing efficient and effective GraphLLM methods for zero-shot graph learning, ultimately pushing the boundaries of graph foundation models.

## 4.3 Time and Memory Efficiency (RQ3)

In this section, we analyze the efficiency and memory cost of representative GraphLLM algorithms on datasets from multiple domains. Figure 3 clearly demonstrates that the current GraphLLM methods generally have higher time and space complexity compared to traditional GNNs. This limitation restricts the application of GraphLLM on large-scale graphs. We can observe that some methods, especially LLM-as-enhancer methods, can achieve relatively good performance improvement with less complexity increase. Besides, although some algorithms (e.g., GLEM [61] and ENGINE [64]) achieve remarkable effectiveness improvement, they largely increase the complexity of time and space. Given these findings, it is important to address the efficiency problem to ensure that GraphLLM methods can be deployed successfully in a wide spectrum of real-world scenarios.

## 5 Conclusion

This paper introduces GLBench, a comprehensive benchmark for the emerging GraphLLM paradigm, by reimplementing and comparing different categories of existing methods across diverse datasets. The fair comparison unearths several key findings on this promising research topic. Firstly, GraphLLM methods achieve better results than traditional GNN- and PLM-based methods in supervised settings. In specific, LLM-as-enhancer methods show the most robust performance across different datasets. Aligning GNNs and LLMs can enhance the effectiveness of both, and the aligned GNNs seem more reliable. However, employing LLMs as predictors has proven to be less effective and tends to have output format issues. Secondly, experimental results indicate that there is no obvious scaling law in existing GraphLLM methods. Thirdly, both structures and semantics are crucial for zero-shot transfer, and we find a simple baseline that can even outperform several existing methods tailored for zero-shot scenarios. Lastly, we highlight the efficiency problem of the existing methods. We believe GLBench will have a positive impact on this emerging research domain. We have made our code publicly available and welcome further contributions of new datasets and methods.

**Acknowledgement.** This work was supported by National Key Research and Development Program of China Grant No. 2023YFF0725100 and Guangzhou-HKUST(GZ) Joint Funding Scheme 2023A03J0673. We want to thank Haitao Mao at Michigan State University, Xinni Zhang and Zhixun Li at The Chinese University of Hong Kong, Wenxuan Huang at Zhejiang University, Ruosong Ye at Rutgers University, and Caiqi Zhang at Cambridge University for their valuable insights into the field of GraphLLM, and constructive comments on this paper.

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

# Appendix

# A  Additional Details on `GLBench`

## A.1  Details of Datasets

All of the public datasets used in our benchmark were previously published, covering a multitude of domains. For each dataset in `GLBench`, we store the graph-type data in the .pt format using PyTorch. This includes shallow embeddings commonly used in classical methods, raw text of nodes, edge indices, node labels, label names, and masks for training, validation, and testing procedures. All datasets are under the MIT License unless otherwise specified. The detailed descriptions of these datasets are listed in the following:

- **Cora.** The Cora dataset is a citation network comprising research papers and their citation relationships within the computer science domain. The raw text data for the Cora dataset was sourced from the GitHub repository provided in Chen et al. [8]. In this dataset, each node represents a research paper, and the raw text feature associated with each node includes the title and abstract of the respective paper. An edge in the Cora dataset signifies a citation relationship between two papers. The label assigned to each node corresponds to the category of the paper.

- **Citeseer.** The Citeseer dataset is a citation network comprising research papers and their citation relationships within the computer science domain. The TAG version of the dataset contains text attributes for 3,186 nodes, and the raw text data for the Citeseer dataset was sourced from the GitHub repository provided in Chen et al. [8]. Each node represents a research paper, and each edge signifies a citation relationship between two papers.

- **Pubmed.** The Pubmed dataset is a citation network that includes research papers and their citation relationships within the biomedical domain. The raw text data for the PubMed dataset was sourced from the GitHub repository provided in Chen et al. [8]. Each node represents a research paper, and each edge signifies a citation relationship between two papers.

- **Ogbn-arxiv.** The Ogbn-arxiv dataset is a citation network comprising papers and their citation relationships, collected from the arXiv platform. The raw text data for the Ogbn-arxiv dataset was sourced from the GitHub repository provided in OFA [29]. The original raw texts are available here[3]. Each node represents a research paper, and each edge denotes a citation relationship.

- **WikiCS.** The WikiCS dataset is an internet link network where each node represents a Wikipedia page, and each edge represents a reference link between pages. The raw text data for the WikiCS dataset was collected from OFA [29]. The raw text associated with each node includes the name and content of a Wikipedia entry. Each node's label corresponds to the category of the entry.

- **Reddit.** The Reddit dataset is a social network where each node represents a user, and edges denote whether two users have replied to each other. The raw text data for the Reddit dataset was collected from GraphAdapter [19]. The raw text associated with each node consists of the content from the user's historically published subreddits, limited to the last three posts per user. Each node is assigned a label that specifies if the user is classified as popular or normal.

- **Instagram.** The Instagram dataset is a social network where nodes represent users and edges represent following relationships. The raw text data for the Instagram dataset was collected from GraphAdapter [19]. The raw text associated with each node includes the personal introduction of this user. Each node is labeled to indicate whether the user is commercial or normal.

## A.2  Details of GraphLLM Models

In our developed `GLBench`, we integrate 12 state-of-the-art GraphLLM methods, including 5 LLM-as-enhancer methods, 5 LLM-as-predictor methods, and 2 LLM-as-aligner methods. We provide detailed descriptions of these methods used in our benchmark as follows.

- **LLM-as-enhancer methods**
  - **GIANT** [10] proposes to conduct neighborhood prediction with the use of XR-Transformers [56] and results in an LLM that can output better feature vectors than bag-of-words and vanilla BERT [23] embedding for node classification.

---

[3]https://snap.stanford.edu/ogb/data/misc/ogbn_arxiv

- **TAPE** [15] uses customized prompts to query LLMs, generating both prediction and text explanation for each node. Then, DeBERTa [14] is fine-tuned to convert the text explanations into node embeddings for GNNs. Finally, GNNs can use a combination of the original text features, explanation features, and prediction features to predict node labels.
- **OFA** [29] describes all nodes and edges using human-readable texts and encodes them from different domains into the same space by LLMs. Subsequently, the framework is adaptive to perform different tasks by inserting task-specific prompting substructures into the input graph.
- **ENGINE** [64] adds a lightweight and tunable G-Ladder module to each layer of the LLM, which uses a message-passing mechanism to integrate structural information. This enables the output of each LLM layer (i.e., token-level representations) to be passed to the corresponding G-Ladder, where the node representations are enhanced and then used for node classification.
- **ZeroG** [26] leverages a language model to encode node attributes and class descriptions, employing prompt-based subgraph sampling and lightweight fine-tuning strategies to address cross-dataset zero-shot transferability challenges in graph learning.

- **LLM-as-predictor methods**
  - **InstructGLM** [55] designs templates to describe local ego-graph structure (maximum 3-hop connection) for each node and conduct instruction tuning for node classification.
  - **GraphText** [62] decouples depth and scope by encapsulating node attributes and relationships in the graph syntax tree and processing it with an LLM.
  - **GraphAdapter** [19] uses a fusion module to combine the structural representations obtained from GNNs with the contextual hidden states of LLMs (e.g., the encoded node text). This enables the structural information from the GNN adapter to complement the textual information from the LLMs, resulting in a fused representation that can be used for supervision training.
  - **GraphGPT** [44] proposes to initially align the graph encoder with natural language semantics through text-graph grounding, and then combine the trained graph encoder with the LLM using a projector. Through a two-stage instruction tuning, the model can directly complete graph tasks with natural language, thus performing zero-shot transferability.
  - **LLaGA** [6] utilizes node-level templates to restructure graph data into organized sequences, which are then mapped into the token embedding space. This allows LLMs to process graph-structured data with enhanced versatility, generalizability, and interpretability.

- **LLM-as-aligner methods**
  - **GLEM** [61] integrates the graph models and LLMs, specifically DeBERTa [14], within a variational Expectation-Maximization (EM) framework. It alternates between updating LLM and GNN in the E-step and M-step, thereby improving effectiveness in downstream tasks.
  - **PATTON** [22] extends GraphFormer [53] by proposing two pre-training strategies, i.e., network-contextualized masked language modeling and masked node prediction, specifically for text-attributed graphs.

## B  Additional Details on Implementations

### B.1  Hyperparameter Tuning

- For GCN, GraphSAGE, and GAT, we grid-search the hyperparameters
```
num_layers in [2, 3, 4], hidden_dim in [64, 128, 256],
dropout in [0.3, 0.5, 0.6].
```

- For Sentence-BERT, BERT, and RoBERTa, we grid-search the hyperparameters
```
lr in [5e-4, 1e-3], dropout in [0, 0.1], batch_size in [4, 8, 16],
label_smoothing in [0, 0.1].
```

- For GIANT, we grid-search the GNN hyperparameters
```
hidden_dim in [64, 128, 256], dropout in [0.3, 0.5, 0.6].
```

- For TAPE, we grid-search the GNN hyperparameters

```
num_layers in [2, 3, 4], hidden_dim in [64, 128, 256],
dropout in [0.3, 0.5, 0.6].
```

- For OFA, we follow the instruction from the original paper [29], use

```
Supervised:
lr=0.0001, 'JK'=none, num_layers=6, hidden_dim=768, dropout=0.15.
Zero-shot:
lr=0.0001, 'JK'=none, num_layers=5, hidden_dim=768, dropout=0.15.
```

- For ENGINE, we follow the instructions from the original paper [64], use

```
hidden_dim=64, dropout=0.5, num_layers=2, activation='elu',
norm='id', sampler='rw', r=32, T=0.1, lr=5e-4, weight_decay=5e-3.
```

- For ZeroG, we grid-search the hyperparameters

```
iterations of neighborhood aggregation in [6, 7, 8, 9, 10],
number of hops for subgraph extraction in [1, 2, 3].
```

- For InstructGLM, we follow the instruction from the original paper [55], use

```
lr=8e-5, warmup_ratio=0.05, dropout=0.1, clip_grad_norm=1.
```

- For GraphText, we follow the instructions from the original paper [62], use

```
lr=0.00005, dropout=0.5, subgraph_size=3, max_length=1024.
```

- For GraphAdapter, we follow the instructions from the original paper [19], use

```
lr=0.0005, num_layers=2, max_length=2048,
hidden_dim=64 for Instagram and hidden_dim=128 for other datasets.
```

- For GraphGPT, we follow the instructions from the original paper [44], use

```
lr=0.002, warmup_ratio=0.03, num_train_epochs=3.
```

- For LLaGA, we follow the instruction from the original paper [6], use

```
lr=2e-3, warmup_ratio=0.03, use_hop=2, sample_neighbor_size=10,
batch_size=16, lr_scheduler_type='cosine', max_length=4096.
```

- For GLEM, we grid-search the GNN hyperparameters

```
num_layers in [2, 3, 4], hidden_dim in [64, 128, 256],
dropout in [0.3, 0.5, 0.6].
```

- For PATTON, we follow the instruction from the original paper [22], use

```
lr=0.00001, warmup_ratio=0.1, dropout=0.1,
batch_size=64, max_length=256.
```

## B.2 Backbone Selection

To compare various GraphLLM methods as fairly as possible, we try to utilize the same GNN and LLM backbones in our implementations. Table 6 shows the GNN and LLM backbones used in the original implementations of GraphLLM methods, as well as those implemented in GLBench.

Table 6: Selection of GNN and LLM Backbones in `GLBench`.

| Role | Method | Predictor | GNN Backbone | | LLM Backbone | |
|---|---|---|---|---|---|---|
| | | | Original | GLBench | Original | GLBench |
| Enhancer | GIANT [10] | GNN | GraphSAGE/RevGAT | GraphSAGE | BERT | Sent-BERT |
| | TAPE [15] | GNN | RevGAT | GraphSAGE | ChatGPT | LLaMA2-7B |
| | OFA [29] | GNN | R-GCN | GraphSAGE | Sent-BERT | Sent-BERT |
| | ENGINE [64] | GNN | GraphSAGE | GraphSAGE | LLaMA2-7B | LLaMA2-7B |
| Predictor | InstructGLM [55] | LLM | - | - | FLAN-T5/LLaMA1-7B | LLaMA2-7B |
| | GraphText [62] | LLM | - | - | ChatGPT/GPT-4 | LLaMA2-7B |
| | GraphAdapter [19] | LLM | GraphSAGE | GraphSAGE | LLaMA2-7B | LLaMA2-7B |
| | LLaGA [6] | LLM | - | - | Vicuna-7B/LLaMA2-7B | LLaMA2-7B |
| Aligner | GLEM [61] | GNN/LLM | GraphSAGE/RevGAT | GraphSAGE | DeBERTa | Sent-BERT |
| | PATTON [22] | LLM | GT | GT | BERT/SciBERT | BERT |

Table 7: Summary of input prompts utilized in training and inference by various LLM-as-predictor methods, illustrated with the *Cora* dataset.

---

**InstructGLM [55]:** Classify the article according to its topic into one of the following categories: [theory, reinforcement learning, genetic algorithms, neural networks, probabilistic methods, case based, rule learning]. Node represents academic paper with a specific topic, link represents a citation between the two papers. Pay attention to the multi-hop link relationship between the nodes. node (<node>,<title>) is featured with its abstract: <abstract>. Which category should (<node>,<title>) be classified as?

---

**GraphText [62]:** Your goal is to perform node classification. You are given the information of each node in a xml format. Using the given information of a node, you need to classify the node to several choices: [<c0>: Rule_Learning, <c1>: Neural_Networks, <c2>: Case_Based, <c3>: Genetic_Algorithms, <c4>: Theory, <c5>: Reinforcement_Learning, <c6>: Probabilistic_Methods]. Remember, your answer should be in the form of the class label.
<information>
    <feature>
        <center_node> <node> </center_node>
        <1st_feature_similarity_graph> <node> </1st_feature_similarity_graph>
    </feature>
</information>

---

**GraphAdapter [19]:** Here is a research paper from the computer science domain, and its title and abstract reads: <raw text>. Question: Based on the title and abstract provided , this paper is on "___" subject. Answer:

---

**GraphGPT [44]:** Given a citation graph: <graph> where the 0th node is the target paper, with the following information: <raw text> Question: Which of the following specific aspect of diabetes research does this paper belong to: 'Rule_Learning', 'Neural_Networks', 'Case_Based', 'Genetic_Algorithms', 'Theory', 'Reinforcement_Learning', 'Probabilistic_Methods'. Directly give the full name of the most likely category of this paper.

---

**LLaGA [6]:** Given a node-centered graph: <graph>, each node represents a paper, we need to classify the center node into 7 classes: Rule_Learning, Neural_Networks, Case_Based, Genetic_Algorithms, Theory, Reinforcement_Learning, Probabilistic_Methods, please tell me which class the center node belongs to?

---

# C Text Prompt Design

## C.1 LLM-as-Predictor Methods

Table 7 presents the input prompts employed by each LLM-as-predictor method. To ensure a fair comparison, we report the results based on the custom prompts crafted in their respective papers.

## C.2 Zero-shot Inference

Table 8 presents the input prompts for zero-shot inference using LLMs. We follow Chen et al. [8] to design prompts for three citation networks, i.e., Cora, Citeseer, and Pubmed, and design similar prompts for other datasets.

# D Discussion of Scaling Law

According to [31], the neural scaling law on graphs "describes how model performance grows with the scale of training, which have two basic forms: **data scaling law** (how the performance changes with training dataset size) and **model scaling law** (how the performance changes with model size)". Since we maintain the dataset splitting in a fair comparison benchmark, we only discuss the model

Table 8: Summary of input prompts for zero-shot inference using LLMs. The prompt format for citation networks follows Chen et al. [8], and similar formats are adapted for other domains.

**Cora:** Paper: <paper_content>. Task: There are following categories: Case_Based, Genetic_Algorithms, Neural_Networks, Probabilistic_Methods, Reinforcement_Learning, Rule_Learning, Theory. Which category does this paper belong to? Output the most possible category of this paper, like 'XX'.

**Citeseer:** Paper: <paper_content>. Task: There are following categories: Agents, ML (Machine Learning), IR (Information Retrieval), DB (Databases), HCI (Human-Computer Interaction), AI (Artificial Intelligence). Which category does this paper belong to? Output the most possible category of this paper, like 'XX'.

**Pubmed:** Paper: <paper_content>. Task: There are following categories: Experimental, Diabetes Mellitus Type 1, Diabetes Mellitus Type 2. Which category does this paper belong to? Output the most possible category of this paper, like 'XX'.

**WikiCS:** Computer Science article: <entity_content>. Task: There are following branches: Computational Linguistics, Databases, Operating Systems, Computer Architecture, Computer Security, Internet Protocols, Computer File Systems, Distributed Computing Architecture, Web Technology, Programming Language Topics. Which branch does this article belong to? Output the most possible branch of this article, like 'XX'.

**Instagram:** Social media post of a user: <user_content>. Task: There are following categories of users: Normal Users, Commercial Users. Which category does the user belong to? Output the most possible category of this user, like 'XX'.

Table 9: OFA and GLEM results on Cora dataset with different layers and parameter sizes of GNNs.

| Models | Layers | Metric | Units per Layer | | | | |
|---|---|---|---|---|---|---|---|
| | | | 128 | 256 | 512 | 768 | 1024 |
| OFA [29] | 4-layer | #Param. | 592769 | 2168577 | 8269313 | 18302209 | 32267265 |
| | | Acc. | 66.10 | 69.15 | 72.05 | 71.13 | **72.97** |
| | 5-layer | #Param. | 691457 | 2562561 | 9843713 | 21843457 | 38561793 |
| | | Acc. | 64.31 | 68.67 | 72.44 | 74.52 | **75.00** |
| | 6-layer | #Param. | 790145 | 2956545 | 11418113 | 25384705 | 44856321 |
| | | Acc. | 64.17 | 69.54 | **75.10** | 73.45 | 74.42 |
| GLEM [61] | 2-layer | #Param. | 66966165 | 67562517 | 68755221 | 69947925 | 71140629 |
| | | Acc. (GNN) | 80.95 | 79.01 | **81.48** | 81.24 | 80.51 |
| | | Acc. (LLM) | 72.68 | 73.66 | 72.87 | 72.05 | 72.53 |
| | 3-layer | #Param. | 67114773 | 68154645 | 71119125 | 75263253 | 80587029 |
| | | Acc. (GNN) | 80.66 | 81.24 | 81.29 | **81.78** | 81.43 |
| | | Acc. (LLM) | 71.47 | 72.92 | 71.71 | 73.07 | 73.07 |
| | 4-layer | #Param. | 67263381 | 68746773 | 73483029 | 80578581 | 90033429 |
| | | Acc. (GNN) | 81.77 | 82.01 | **82.16** | 81.53 | 82.06 |
| | | Acc. (LLM) | 72.73 | 71.47 | 71.95 | 72.01 | 72.20 |

scaling law here. To be specific, we select OFA [29] and GLEM [61], which perform well across all datasets in GLBench, for our scaling law experiments. Since these models typically comprise both a **GNN** and an **LLM**, we explore scaling law separately for each component. To ensure a fair comparison, we adopt consistent experimental setups across different scales of GNNs and LLMs, such as "learning_rate" and "batch_size" (more details on parameters can be found in Appendix B.1), with the only variable being the scales of GNNs and LLMs.

### D.1 Different scales of GNNs

We utilize fixed LLMs (i.e., SentenceBERT) and explore the scaling laws of GNNs in GraphLLM methods using GNNs with different layers and dimensions of hidden states. Detailed results for OFA and GLEM are shown in Table 9 and Figure 4, respectively. From the results, we observe that as the number of parameters increases, whether due to an increase in layers or the dimensions, there is no significant trend of improvement in model performance.

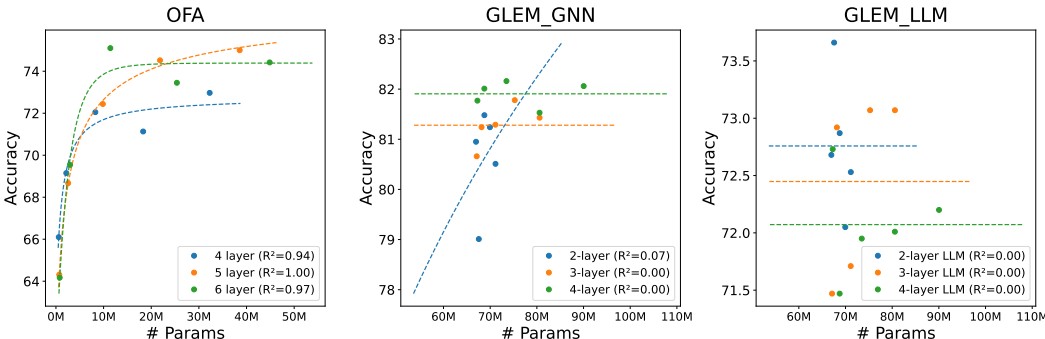

Figure 4: Model scaling behaviors of OFA and GLEM with varying model depths. $R^2$ close to 1 indicates that the model has a strong explanatory power. $R^2$ close to 0 means that the model explains or predicts 0% of the relationship between the dependent and independent variables.

Table 10: OFA and GLEM results on Cora dataset with different parameter sizes of LLMs.

| Method | TinyBERT 4M | | BERT-base 109M | | BERT-large 335M | | Sent-BERT_small 23M | | Sent-BERT 66M | | RoBERTa_base 125M | | RoBERTa_large 355M | | LLaMA2-7B 7B | | LLaMA2-13B 13B | |
|---|---|---|---|---|---|---|---|---|---|---|---|---|---|---|---|---|---|---|
| | GNN | LLM | GNN | LLM | GNN | LLM | GNN | LLM | GNN | LLM | GNN | LLM | GNN | LLM | GNN | LLM | GNN | LLM |
| OFA | 56.33 | - | 65.52 | - | 65.18 | - | 78.97 | - | 75.24 | - | 56.00 | - | 58.03 | - | 78.72 | - | **80.56** | - |
| GLEM | 72.15 | 37.52 | **81.57** | 73.11 | 79.06 | 70.60 | 78.00 | 69.05 | 81.23 | 72.92 | 76.74 | 70.89 | 79.26 | 72.53 | OOM | OOM | OOM | OOM |

## D.2 Different scales of LLMs

We use fixed GNNs and explore the scaling laws of LLMs in GraphLLM methods using LLMs ranging from 4M to 13B. The results are shown in Table 10. For fairness, we introduce several versions of the same LLM-series for comparison, such as BERT-base and BERT-large. It can be observed that increasing the scale of LLMs has a less noticeable impact on model performance. Although OFA with LLaMA2-13B achieves the best results, the performance does not increase steadily with the increase in parameters, e.g., OFA with RoBERTa-large only achieves 58.3% Accuracy.

In summary, it is hard to identify clear scaling laws, whether related to the scaling of GNNs or LLMs, which is consistent with the insights in GLBench.

# E Broader Discussion

## E.1 Limitations of `GLBench`

In `GLBench`, we only consider the *node classification* task, due to the fact that most GraphLLM methods only support a single task and node classification is the task most methods focus on. With the development of Graph Foundation Models [35], tasks can be extended to *edge-level* and *graph-level* as many methods such as OFA [29], LLaGA [6], and UniAug [45] already support multi-task pre-training and inference. In addition, the absence of non-text-attributed graphs is also a concern, as many real-world graphs lack textual information [25].

## E.2 Potential Impacts

Leveraging LLMs to assist graph tasks is a fast-growing and promising research field, covering a wide range of applications. We introduce this benchmark to call more researchers' attention to this new paradigm, i.e., GraphLLM. The proposed benchmark `GLBench` can significantly facilitate the development of this community. Specifically, `GLBench` deeply and extensively explores the paradigm of combining LLMs into graph tasks, including three categories of methods: LLM-as-enhancer, LLM-as-predictor, and LLM-as-aligner. It provides a comprehensive evaluation of datasets across different domains. In the future, we will keep track of newly emerged techniques in the GraphLLM field and continuously update `GLBench` with more solid experimental results and detailed analyses.

