# OpenReview forum: "GLBench: A Comprehensive Benchmark for Graph with Large Language Models"
_NeurIPS.cc/2024/Datasets_and_Benchmarks_Track — NeurIPS 2024 Track Datasets and Benchmarks Poster_

### Official Review · Reviewer_rfpz · 2024-07-05
**Reviews**

**Rating:** 7
**Confidence:** 4
**Clarity:** Yes, the paper is well written.

**Review:**

See below.

**Strengths:**

This paper is the first to propose a comprehensive benchmark for evaluating GraphLLM methods, encompassing a diverse range of approaches, including GNN-based methods, PLM-based methods, and various roles played by LLMs. The benchmark is general and representative, providing a robust framework for evaluation. Additionally, the paper conducts experiments in both supervised and zero-shot scenarios, and evaluates training time and space efficiency.

The insights offered are significant and valuable, particularly in understanding how different roles played by LLMs influence overall performance. These findings could guide the future integration of GNNs and LLMs. Moreover, the paper is well-written and easy to follow, facilitating further improvements and subsequent research in the field.

**Additional Feedback:**

I hope the authors can provide a clearer discussion on how this benchmark relates to other relevant works and optimize their repository. Additionally, incorporating the suggested experiments listed in the "Opportunities for Improvement" section would be highly appreciated.

**Correctness:**

The evaluation methods and experiment design are appropriate and performed correctly.

**Documentation:**

I have checked the URL provided in the paper and have the following suggestions:

1. The models used in this paper have been included, but they are not well-aligned. Some baselines appear to be directly cloned from their original repositories, resulting in inconsistent organization. Improved documentation would greatly enhance reproducibility.
2. The datasets are not comprehensively included, resulting in an incomplete collection. It is recommended to add the missing datasets along with their generation and splitting codes.
3. The repository does not include the experimental codes, which are expected to be provided.

**Limitations:**

The authors have adequately addressed the limitations of their work.

**Opportunities For Improvement:**

* While the benchmark is general enough to provide basic performance comparisons between different methods, it falls short of delving deeply into the specific challenges each method addresses. For instance, in supervised settings, it would be valuable to explore how imbalance issues dynamically affect model performance at varying levels, such as from 10% to 50%. Additionally, examining how transferability varies with different pre-training datasets and settings would enhance the benchmark’s representativeness. This deeper analysis would provide more nuanced insights.

* Some experimental conclusions may require additional results or comparisons for greater rigor. A notable example is conclusion 5 (There is no obvious scaling law in existing GraphLLM methods). To substantiate this claim, more rigorous experiments are necessary. For instance, maintaining constant pre-train datasets, models, downstream datasets, and the role of LLMs while only varying the model size would provide clearer insights into performance improvements. Repeating these experiments across different models and datasets would ensure more representative and reliable findings. The related experiments in this paper lack detailed design, which could result in less conclusive insights.

**Relation To Prior Work:**

The paper does not clearly discuss its relation to prior work or previous contributions within the main text. Although it is true that there are few works focused on GraphLLM benchmarks, the authors should still address the relationships with relevant works, such as other graph benchmarks.

**Summary And Contributions:**

The paper presents GLBench, a comprehensive benchmark designed to evaluate GraphLLM methods. GLBench aims to standardize experimental settings, ensuring fair and thorough evaluations across diverse real-world datasets. It provides insights into both supervised and zero-shot scenarios while analyzing the training time and space efficiency of various GraphLLM methods.

The contributions of this paper can be summarized as follows:

- It ensures fair evaluations across real-world datasets, facilitating direct comparisons between existing models.
- It offers significant insights into GraphLLM and LLM-based methods, enhancing the understanding of the GraphLLM field and paving the way for advancements in graph foundation models.
- GLBench encompasses experiments in both supervised and zero-shot scenarios using multiple evaluation metrics, providing a comprehensive analysis of various models.

---

> ### Author Rebuttal · Authors · 2024-08-17
>
> We greatly appreciate you for the insightful review to help us improve the paper. We address the concerns as follows:
>
> > W1: How imbalance issues affect model performance.
>
> Thank you for your insightful suggestions. Node imbalance is a crucial task for GNNs, as they often exhibit a bias toward majority classes. Investigating whether GraphLLM methods can offer more stable performance in such settings is highly meaningful. However, as a benchmark **designed to evaluate the effectiveness of GraphLLM methods, we prefer to conduct experiments under stable, class-balanced conditions**. For instance, the data splits we use for five datasets are balanced. We hope that with the ongoing development of GraphLLM methods, such issues will be rigorously investigated in future research.
>
> > W2: How transferability varies with different pre-training datasets and settings.
>
> We thank the reviewer for raising this novel perspective! In GLBench we specifically use the two biggest graphs, Ogbn-arxiv and Reddit, as our pre-training datasets for transferability experiments. Based on your suggestions, two new questions have arisen: **1) Is more pre-training data always better? 2) Does cross-domain pre-training significantly affect transferability?** To answer these questions, we expand our transfer experiments, using Cora as the downstream dataset, and report the performance of OFA and ZeroG based on different pre-training datasets. We evaluate OFA and ZeroG performance across various pre-training datasets, analyzing results in both in-domain and cross-domain settings. The results are shown in `Table 1` of the attached PDF.
>
> From the table we can observe that: **1) Simply increasing pre-training data does not necessarily lead to stable improvements, and may even result in performance degradation.** For example, using Arxiv+Citeseer+Pubmed instead of just Arxiv for pre-training decreases the zero-shot transferability of OFA and ZeroG on the Cora dataset by 5.03% and 0.24% respectively. This may be attributed to negative transfer. **2) Compared to in-domain pre-training, cross-domain pre-training significantly decreases the performance on downstream datasets.** We observe that while OFA reaches a maximum accuracy of 44.87% in in-domain pre-training, it drops to 12.74% in cross-domain pre-training, with a 32.13% accuracy decrease. This demonstrates that semantic and structural differences between domains significantly impact the transferability of existing GraphLLM methods.
>
> In summary, discussing the impact of different pre-training datasets on transferability is fascinating. Researchers can conveniently utilize our benchmark to design pre-training and inference datasets to evaluate their model's transferability.
>
> > W3: Lack of detailed design in scaling law.
>
> Thanks for pointing out this issue! Based on your comment and Reviewer eYMz's W3, we acknowledge our past misunderstandings of scaling laws and flawed experimental design within GLBench. It is unreasonable to analyze scaling law by directly comparing different GraphLLM methods. We have now designed a more **rigorous and accurate** experiment to better analyze the scaling laws here, and we are committed to incorporate these results and conclusions in the next version of our paper :)
>
> According to [1], the neural scaling law on graphs "describes how model performance grows with the scale of training, which have two basic forms: **data scaling law** (how the performance changes with training dataset size) and **model scaling law** (how the performance changes with model size)." Since we maintain the dataset splitting in a fair comparison benchmark, we only discuss the model scaling law here. To be specific, we select OFA [ICLR'24] and GLEM [ICLR'23], which perform well across all datasets in GLBench, for our scaling law experiments. Since these models typically comprise both a **GNN** and an **LLM**, we explore scaling law separately for each component. To ensure a fair comparison, we adopt consistent experimental setups across different scales of GNNs/LLMs, such as `learning_rate` and `batch_size` (more details on parameters can be found in Appendix C.1), with the only variable being the scales of GNNs/LLMs.
>
> - **Different scales of GNNs**: We utilize fixed LLMs (i.e., SentenceBERT) and explore the scaling laws of GNNs in GraphLLM methods using GNNs **with different layers and dimensions of hidden states**. Detailed results for OFA and GLEM are shown in `Table 2` and `Table 3` of the attached PDF, respectively. From the results, we observe that as the number of parameters increases, whether due to an increase in layers or an increase in the dimensions, there is no significant trend of improvement in model performance. This observation is more evident in the fitting of `Figure 1` of the attached PDF.
>
> - **Different scales of LLMs**: We use fixed GNNs and explore the scaling laws of LLMs in GraphLLM methods **using PLMs ranging from 4M to 13B**. The results are shown in `Table 4` of the attached PDF. For fairness, we introduce several versions of the same LLM-series for comparison, such as BERT-base and BERT-large. We find that increasing the scale of LLMs has less noticeable impact on model performance. Although OFA with LLaMA2-13B achieves the best results, the performance does not increase steadily with the increase in parameters, e.g., OFA with RoBERTa-large only achieves 58.3% ACC.
>
> In summary, it is hard to identify clear scaling laws, whether related to the scaling of GNNs or LLMs, which is consistent with Insight 4 in GLBench.
>
> *References*:
>
> [1] Neural Scaling Laws on Graphs. *Arxiv'24*
>
> > W4: Docs-related: Improve documentation, missing datasets, and experimental code.
>
> Thanks for your kind reminder! We have taken your suggestions into consideration and believe the current Repo is well organized. GLBench can be found at https://github.com/NineAbyss/GLBench, we hope you can check it and would greatly appreciate any further suggestions you might have.

---

> > ### Comment · Reviewer_rfpz · 2024-08-19
> >
> > Thanks for the authors' efforts in providing more results to address my concerns, especially the discussion of scaling law. The authors are suggested to add these new findings in the paper accordingly. Moreover, I encourage the authors to delve deeper into the research on the scaling law and models' performance under other conditions (W1). With the new results, the experiments would be more comprehensive and convincing.  I will raise the rating accordingly.

---

> > > ### Author Response · Authors · 2024-08-19
> > >
> > > Thanks for your constructive comments and for raising your score! We will include more results in our revised paper to better support our evaluation of scaling law.

---

### Official Review · Reviewer_1oX9 · 2024-07-26

**Rating:** 6
**Confidence:** 4
**Correctness:** Mostly correct.
**Clarity:** Yes

**Review:**

Pros
1. This paper unifies the evaluation protocols of several benchmark datasets, including Cora, Citeseer, Pubmed, Ogbn-arxiv. Therefore, we can have a fair comparison for different methods.
2. From the experimental analysis, we could learn some interesting findings of GraphLLMs, such as "There is no obvious scaling law in existing GraphLLM methods"
3. The writing is clear and easy to follow in general.


Cons
1. This paper does not compare with Self-Supervised Learning (SSL) methods for the supervised setting. I would like to see how GraphLLMs perform compared with SSL methods.
2. Some details need further explanation.
- How do you use SSL methods for zero-shot learning setting?
- For the "Training-free" method in Table 4, which LLM do you use?
- Can you provide further reasons and findings that why the "training-free" method outperform GraphLLMs such as GraphGPT, ZEROG and OFA?
3. Some writing suggestions. Maybe you can merge section 3&4 for better readability.

**Strengths:**

See pros.

**Additional Feedback:**

N/A

**Documentation:**

Yes

**Opportunities For Improvement:**

See cons.

**Relation To Prior Work:**

Yes

**Summary And Contributions:**

This paper introduces GLBench, which is a comprehensive benchmark for evaluating GraphLLM methods in both supervised and zero-shot settings. On the proposed benchmark, this paper compares 3 groups of methods, including traditional Graph Neural Networks (GNNs), Pretrained Language Models (PLM) and GraphLLMs.

---

> ### Author Rebuttal · Authors · 2024-08-17
>
> Thank you very much for the careful reading and comments regarding our work. We address your concerns as follows.
>
> > Q1: Compare with GraphSSL methods in supervised settings.
>
> Thank you for highlighting the need for add GraphSSL methods! GraphSSL has attracted significant attention for its ability to utilize large amounts of unlabeled samples in graphs and has achieved better results compared to traditional GNNs. We have included GraphSSL methods in our comparison with GraphLLM methods in a supervised setting, and we are committed to incorporate these results and conclusions in the next version of our paper :)
>
> Specifically, we adopt three commonly used GraphSSL methods, i.e.,  **DGI** (contrastive-based) [ICLR'18], **GraphMAE** (generative-based) [SIGKDD'21], and **CCA-SSG** (contrastive-based) [NeurIPS'21] to conduct experiments. For a fair comparison, we use the same **SentenceBert embedding** as the initial node feature, employ the same **data splitting**, and utilize the same **hyperparameter space for grid search** (details can be found in Appendix C.1). We leverage GCN as the backbone for GraphSSL methods. The comparison between GraphSSL methods and other methods can be found in `Table 1` of the attached PDF.
>
> From `Table 1` we can observe that: 1) **Compared to GNN methods**: GraphSSL methods demonstrate significant improvements on most datasets, notably Cora and Reddit, with accuracy gains of approximately 2% and 7% respectively, showing the effectiveness of GraphSSL. 2) **Compared to GraphLLM methods**: While GraphSSL enhances GNN-based performance, GraphLLM methods still outperform them on five datasets. For example, on Citeseer, Arxiv, and WikiCS, the best GraphLLM models outperform the top-performing GraphSSL-based models by 2.19%, 5.91%, and 3.57% in accuracy, respectively. This suggests GraphLLMs better leverage both structure and semantics, enabling more powerful graph learning than traditional GNNs, even when enhanced by self-supervised learning.
>
> > Q2: How do you use SSL methods for zero-shot learning setting?
>
> Thanks for your question. To provide a clear explanation, we also visually illustrate this process in `Figure 1` of the attached PDF. Specifically,
>
> **GraphSSL Pre-training Stage:**
>
> - Step (a): We employ an LLM to obtain the initial features of nodes.
> - Step (b): A GNN is pre-trained using GraphSSL techniques based on structures of pre-training graphs.
>
> **Zero-shot Inference Stage:**
>
> - Step (a): Given a target graph, we initially apply the same LLM used during pre-training to generate initial embeddings for both nodes and classes.
> - Step (b): We then apply the pre-trained GNN to the target graph to update the node embeddings. Subsequently, we compute the dot product similarities between node and class embeddings, and the class that yields the highest similarity score is predicted to be the class of the node.
>
>
> > Q3: For the "Training-free" method in Table 4, which LLM do you use?
>
> Thanks for your question. To ensure a fair comparison with baselines, consistent with OFA and ZeroG, we use **SentenceBERT** as the LLM backbone in the "Training-free" method, specifically `sentence-transformers/multi-qa-distilbert-cos-v1` version. Inspired by your question, we are curious about the impact of different LLMs on the "training-free" method. We evaluate different LLMs in "training-free" method with different architectures and sizes of parameters, including encoder-only LLMs such as **BERT** and **RoBERTa**, encoder-decoder LLMs such as **T5**, and decoder-only LLMs such as **LLaMA2**. The performance is shown in `Table 2` in the attached PDF.
>
> We observe some new insights based on the results. 1) Auto-regressive LLMs such as T5 and LLaMA are specifically tuned for conversational purposes, and **not perform as well in sentence encoding scenarios**. 2) Compared to BERT and RoBERTa, **both versions of SentenceBERT can effectively capture semantics and generate higher quality sentence embeddings**. We think this is also why both OFA and ZeroG use SentenceBERT as LLMs in their papers.

---

> > ### Author Response · Authors · 2024-08-17
> >
> > > Q4: Can you provide further reasons and findings that why the "training-free" method outperform GraphLLMs such as GraphGPT, ZEROG and OFA?
> >
> > Thanks for raising this interesting question. We were also surprised to find that our "training-free" method can even outperform several GraphLLM methods tailored for zero-shot scenarios that require pre-training. Upon careful analysis, we gradually realize that **it's not that our method is too good, but rather that the transferability of existing methods is too poor**. For example, on Cora with 7-class, OFA and GraphGPT **only** achieve accuracy of 23.11% and 24.9% respectively, which is not much better than random guessing at 14.3% accuracy. In other words, the pre-training stage is almost useless in these methods. We find the following reasons led to the negative transfer:
> >
> > 1. **Mismatched classification head**. In pre-training, existing methods usually need to learn a classification head (typically a linear layer $\in$ $\mathbb{R}^{dim \times class}$) to classify nodes. However, **pre-training and downstream datasets usually contain different numbers of classes. Even when the number of labels is the same, labels may carry different meanings across datasets**. For example, the categories in citation networks may not easily translate to those in web link datasets due to the distinct contexts they represent. This leads to poor performance of the classification head learned from pre-training in downstream datasets, which is a problem OFA [ICLR'24] encountered. On the other hand, the "training-free" method avoids this issue as it has no classification head.
> > 2. **Overfitting issues**. In graph data, negative transfer often occurs when there is a significant disparity in structure or semantics between graphs. **Fully adapting GraphLLM methods to pre-training datasets often causes overfitting, where the model becomes too specialized to the pre-training data’s characteristics**. This may hurt the model’s performance on target datasets that differ in structure or meaning, as it may fail to capture the broader patterns necessary for zero-shot transfer learning. For an example in GraphGPT [SIGIR'24], when performing inference on WikiCS, the answers provided are all based on citation network seen during pre-training.
> >
> > >     {
> > >         "id": "wikics_test_5836",
> > >         "node_idx": 5836,
> > >         "res": "Based on the information provided, the citation graph is a directed graph where the 0th node represents the target paper on diabetes research. The graph contains information about block contention, which refers to multiple processes competing for access to the same data or resources. The goal is to avoid contention situations,"
> > >     }
> >
> > 3. **Lack of semantic transfer**. GraphLLM methods typically freeze LLM parameters during pre-training and only update GNN parameters, like OFA [ICLR'24] does. This means that **the semantic spaces of pre-training and downstream datasets are never aligned**, lacking semantic transfer. This seems unreasonable, especially when the pre-training and downstream domains are the same. For instance, intuitively, pre-training on the ogbn-arxiv's corpus should help with downstream inference on Cora.
> >
> > In summary, existing GraphLLM methods still cannot effectively transfer knowledge, both semantically and structurally. We look forward to the development of better Graph Foundation Models with zero-shot capabilities, achieving true positive transfer.
> >
> > > Q5: Writing suggestions: merge section 3&4 for better readability.
> >
> > Thanks for your suggestions! We agree that due to the strong correlation between Section 3&4, where each research question in Section 3 corresponds to an experimental result in Section 4, merging Sections 3&4 will enhance coherence and readability. We will make this change in the next version, thank you.

---

### Official Review · Reviewer_n5o4 · 2024-07-26
**Graph-LLM benchmarks**

**Rating:** 8
**Confidence:** 4
**Correctness:** Yes.
**Clarity:** Yes.

**Review:**

It is a nice and timely paper, and most likely will be cited, used by the community.

**Strengths:**

Very well structured and easy to read.
The code is packaged nicely and is easy to use.

**Additional Feedback:**

NA

**Documentation:**

Yes

**Limitations:**

No, I do not find this explicitly discussed.

**Opportunities For Improvement:**

Better explaining what is lacking in the evaluation of some of the recent papers, and the need for an independent benchmarking framework.

Including more datasets, and more tasks.

**Relation To Prior Work:**

Well positioned., although some works are missing, e.g.
https://openreview.net/forum?id=IuXR1CCrSi

**Summary And Contributions:**

The authors propose benchmarks for evaluating recent graph+LLM models. They unify the experiment setting, splits, for seven datasets used in different papers, and provide a comparison on supervised and zero shot tasks for a comprehensive list of models. Several findings are reported which are interesting on their own, for example on how scaling doesn't help the performance of these models and how their training time compares with the classic models.

---

> ### Author Rebuttal · Authors · 2024-08-17
>
> We sincerely appreciate your recognition of our paper and your valuable time. We address the concerns as follows:
>
> > W1: Better explaining what is lacking in the evaluation of some of the recent papers, and the need for an independent benchmarking framework.
>
> Thank you for your valuable suggestions! The main motivation we developed GLBench is due to some observed issues in some of the recent papers, which include:
>
> - **Unfair comparisons:** This can be primarily divided into two categories: **1) Inconsistent data splitting**. Most GraphLLM methods still follow Yang et al. [ICLR'16] to conduct experiments on the Cora, Pubmed, and Citeseer with fixed 20 labeled training nodes per class (i.e., *20-shot*). However, we notice that several works like TAPE [ICLR'24], InstructGLM [EACL'24], and GraphGPT [SIGIR'24] employ a *60%/20%/20% train/validation/test split*, using far more nodes for training and making it challenging to fairly compare with other methods. **2) Inconsistent text attributes**. Give the same node, different methods construct its text attributes in different ways. Taking Cora as an example, most methods use *the concatenation of the paper's title and abstract* as its text attributes. However, TAPE [ICLR'24] additionally uses ChatGPT to further retrieve *additional knowledge*, improving the length and quality of its text attributes, leading to unfair comparisons.
> - **Incomplete comparisons:** Existing GraphLLM methods mostly compare only a few datasets within a single domain, *lacking comprehensive discussions on text-attributed-graphs of different scales, domains, and densities*. For example, GraphAdapter [WWW'24] focuses on two social networks, while TAPE [ICLR'24], InstructGLM [EACL'24], and GraphGPT [SIGIR'24] only conduct experiments on citation networks. We recognize the need for a benchmark to explore the comprehensive performance of existing methods across multiple datasets.
> - **Lack of unified evaluation for zero-shot transferability：** Although several GraphLLM methods have this capability, there is currently no unified benchmark to fairly compare their zero-shot transferability. For instance, using Cora for downstream inference, GraphGPT [SIGIR'24] uses *ogbn-Arxiv and Pubmed* for pre-training, ZeroG [SIGKDD'24] uses *ogbn-Arxiv, Pubmed, and Citeseer* for pre-training, while OFA uses *ogbn-Arxiv, FB15K237, and Chemble* for pre-training. It's difficult for researchers to intuitively compare the zero-shot performance of existing works.
> - **Lack of unified discussion on computation and memory usage：** Different categories of GraphLLM method significantly vary in efficiency and memory usage. For example, LLM-as-enhancer generally operates faster, while LLM-as-predictor tends to be slower. *Understanding this trade-off between efficiency and performance is essential for researchers choosing appropriate GraphLLM methods for practical applications.*
>
> Motivated by these issues, we aim to provide fairer evaluations to this community through GLBench, enhancing clarity and understanding for researchers. We are willing to revise the Introduction in the final of the paper to clarify our motivation.
>
> > W2: Including more datasets, and more tasks.
>
> Thank you for your advice! We acknowledge that discussing a broader range of datasets and tasks would be beneficial. Additionally, we have explored two new tasks: 1) The performance of GraphSSL methods compared to GraphLLM methods; 2) The impact of different pre-training datasets on zero-shot transferability.
>
> - **GraphSSL vs. GraphLLM**
>
> Specifically, we adopt three commonly used GraphSSL methods in the supervised setting, i.e.,  **DGI** (contrastive-based)[ICLR'18], **GraphMAE** (generative-based)[SIGKDD'21], and **CCA-SSG** (contrastive-based)[NeurIPS'21] to conduct experiments. The comparison between GraphSSL methods and other methods can be found in `Table 1` of the attached PDF. We can observe that: 1) **Compared to GNN methods**: GraphSSL methods demonstrate significant improvements on most datasets. 2) **Compared to GraphLLM methods**: While GraphSSL enhances GNN-based performance, GraphLLM methods still outperform them on five datasets. This suggests GraphLLMs better leverage both structure and semantics, enabling more powerful graph learning than traditional GNNs, even when enhanced by self-supervised learning.
>
> - **Impact of different pre-training datasets**
>
> In GLBench, we use the two biggest graphs as our pre-training datasets for transferability experiments to align with data distribution practices in NLP and CV. Additionally, two interesting questions have arisen: **1) Is more pre-training data always better? 2) Does cross-domain pre-training significantly affect transferability?** To answer these questions, we expand our transfer experiments, using Cora as the downstream dataset, and report the performance of OFA and ZeroG based on different pre-training datasets. The results are shown in `Table 2` of the attached PDF. From the table we can observe that: **1) Simply increasing pre-training data does not necessarily lead to stable improvements, and may even result in performance degradation.** For example, using Arxiv+Citeseer+Pubmed instead of just Arxiv for pre-training decreases the zero-shot transferability of OFA and ZeroG on the Cora dataset by 5.03% and 0.24% respectively. **2) Compared to in-domain pre-training, cross-domain pre-training significantly decreases the performance on downstream datasets.** This demonstrates that semantic and structural differences between domains significantly impact the transferability of existing GraphLLM methods.

---

### Official Review · Reviewer_eYMz · 2024-07-31
**A good paper with solid**

**Rating:** 6
**Confidence:** 4
**Correctness:** The claim and evaluation methods are …
**Clarity:** The presentation is clear, including …

**Review:**

### Pros

1. In general, the idea to design this graph learning benchmark is interesting and and it is timely work.

2. The research questions this benchmark try to figure out are important for the graph community and the systematic analysis revealing interesting insights.

3. The authors make the code available, which can help upcoming research.

### Cons

1. The selected graphs all include nodes with textual features, however, it will be more general if this work can include graphs whose nodes have no textual features, thus the benchmark can be more valuable for the community.

2. Despite various experimental observations, the analysis is limited, which leaves rich questions to be answered.

3. Especially, the scaling law w.r.t GraphLLM in this work is kind of simple and naive. The compared models with different parameters are of different structures, thus the results are not that convincing. It will be more interesting if the authors can discuss more on this topic since the benchmark is for GraphLLM.

**Strengths:**

Check the pros and contributions in the above sections.

**Additional Feedback:**

NA.

**Documentation:**

The documentation including the datasets, code, and experimental details, is clear and comprehensive.

**Ethics:**

No ethical concerns.

**Limitations:**

NA.

**Opportunities For Improvement:**

Give a more concrete design in terms of scaling laws w.r.t the graph domain, which should be different from those in NLP domain.

**Relation To Prior Work:**

The relation to prior work is clear.

**Summary And Contributions:**

### Summary
The paper presents GLBench, a comprehensive benchmark tailored for evaluating GraphLLM methods under supervised and zero-shot learning environments. It addresses a critical gap in the field by providing a unified platform for comparing GraphLLM approaches, including traditional graph neural networks, against a backdrop of diverse and real-world datasets. The benchmark's systematic analysis uncovers that GraphLLM methods, particularly those using LLMs as enhancers, generally surpass traditional methods in supervised tasks. However, the use of LLMs as predictors falls short, often resulting in uncontrollable outputs. The study also dispels the myth of a direct correlation between model size and performance, emphasizing the absence of a clear scaling law. Furthermore, it underscores the importance of both structural and semantic information in zero-shot learning and introduces a surprisingly effective, training-free baseline. The paper concludes with an emphasis on the need for efficiency in GraphLLM methods, noting their higher time and space complexity compared to GNNs

### Contributions

1. The paper introduces GLBench, the first comprehensive benchmark for assessing GraphLLM methods across supervised and zero-shot learning scenarios, providing a unified experimental framework for fair comparisons.

2. It offers a systematic evaluation of existing GraphLLM methods, revealing their performance in various dimensions—supervised effectiveness, zero-shot transferability, and efficiency in terms of computation and memory usage, uncovering the lack of a clear scaling law and the importance of both structure and semantics.

3. By making the benchmark's data, code, and findings publicly accessible, the paper supports and encourages further research in the domain of GraphLLM and graph foundation models, highlighting the need for efficient method deployment in practical applications.

---

> ### Author Rebuttal · Authors · 2024-08-17
>
> Thanks for your insightful suggestions on our paper and valuable time! We try our best to address your concerns as follows.
>
> > W1: It will be more general if this work can include graphs whose nodes have no textual features.
>
> Thanks for your insightful questions! Utilizing LLMs to assist learning on text-attributed graphs (TAGs) has already demonstrated excellent performance. However, graph-structured data is ubiquitous in real-world scenarios, and a great deal of it lacks rich textual information, presenting a challenge for GraphLLM methods when handling non-text-attributed graphs (non-TAGs).
>
> To address this, we can convert a non-TAG into a TAG using a **template-based method that describes the semantic meaning of each node in human-understandable language**, thus enabling GraphLLM methods to handle it. For example, consider the *MovieLens-1m* dataset, a recommendation graph with user and movie nodes where the objective is to predict ratings between them. For movie nodes, we can use the movie's title and relevant descriptions as text attributes. However, user nodes do not have clear text attributes. In this case, we can use a template-based method to **transform user information columns into understandable natural language**. Here are examples of personal information and the template:
>
> |Name| Gender|Age|Occupation|
> |-|-|-| -|
> |Jack|Man|35|Microsoft|
>
> >     User in the movie rating platform with the following information:
> >     gender: <gender>, age: <age>, occupation: <occupation>.
>
> By assigning text attributes to nodes, we enable GraphLLM methods to process them. As your suggestions, we are willing extend our evaluations of non-TAGs in future work to enhance the diversity and impact of GLBench.
>
> > W2: More questions and insights should be discussed.
>
> Thank you for highlighting the need for a more comprehensive analysis. In GLBench, our analysis primarily stems from two parts：1）In supervised scenarios, we compare GraphLLM methods with GNN-based and LLM-based methods (**Insight 1**), analyze three types of GraphLLM methods respectively (**Insights 2-4**), and discuss scaling laws (**Insight 5**). 2）In zero-shot scenarios, we compare the transferability of existing GraphLLM methods with LLM-based methods (**Insight 6**), discuss structural and semantic information in transfer (**Insight 7**), and introduce a superior d"training-free" method (**Insight 8**). We agree that more questions and insights about GraphLLM should be discussed. Therefore, we further explore an interesting question in the zero-shot scenario: **How transferability varies with different pre-training datasets and settings?**.
>
> In GLBench, we specifically use the two biggest graphs, Ogbn-arxiv and Reddit, as our pre-training datasets for transferability experiments to align with data distribution practices in NLP and CV, where source datasets are typically larger than target datasets. However, two new questions have arisen: **1) Is more pre-training data always better? 2) Does cross-domain pre-training significantly affect transferability?** To answer these questions, we expand our transfer experiments, using Cora as the downstream dataset, and report the performance of OFA and ZeroG based on different pre-training datasets. We evaluate OFA and ZeroG performance across various pre-training datasets, analyzing results in both in-domain and cross-domain settings. The results are shown in `Table 1` in the attached PDF.
>
> From the table we can obtain **two new insights** in GLBench: **1) Simply increasing pre-training data does not necessarily lead to stable improvements, and may even result in performance degradation.** For example, using Arxiv+Citeseer+Pubmed instead of just Arxiv for pre-training decreases the zero-shot transferability of OFA and ZeroG on the Cora dataset by 5.03% and 0.24% respectively. This may be attributed to negative transfer. Even in-domain transfer ensures some semantic similarity, the structural differences between graphs, such as density and scale, can still affect transfer performance. **2) Compared to in-domain pre-training, cross-domain pre-training significantly decreases the performance on downstream datasets.** We observe that while OFA reaches a maximum accuracy of 44.87% in in-domain pre-training, it drops to 12.74% in cross-domain pre-training, with a 32.13% accuracy decrease. Although ZeroG appears to have stable performance under cross-domain pre-training (i.e., 59.91%~61.17%), there is still an average decrease of about 3% in accuracy.

---

> > ### Author Response · Authors · 2024-08-17
> >
> > > W3: Lack of detailed design in scaling law.
> >
> > Thanks for pointing out this issue! Based on your comment and Reviewer rfpz's W2, we acknowledge our past misunderstandings of scaling laws and flawed experimental design within GLBench. It is unreasonable to analyze scaling law by directly comparing different GraphLLM methods. We have now designed a more **rigorous and accurate** experiment to better analyze the scaling laws here, and we are committed to incorporate these results and conclusions in the next version of our paper :)
> >
> > According to [1], the neural scaling law on graphs "describes how model performance grows with the scale of training, which have two basic forms: **data scaling law** (how the performance changes with training dataset size) and **model scaling law** (how the performance changes with model size)." Since we maintain the dataset splitting in a fair comparison benchmark, we only discuss the model scaling law here. To be specific, we select OFA [ICLR'24] and GLEM [ICLR'23], which perform well across all datasets in GLBench, for our scaling law experiments. Since these models typically comprise both a **GNN** and an **LLM**, we explore scaling law separately for each component. To ensure a fair comparison, we adopt consistent experimental setups across different scales of GNNs/LLMs, such as `learning_rate` and `batch_size` (more details on parameters can be found in Appendix C.1), with the only variable being the scales of GNNs/LLMs.
> >
> > - **Different scales of GNNs**: We utilize fixed LLMs (i.e., SentenceBERT) and explore the scaling laws of GNNs in GraphLLM methods using GNNs **with different layers and dimensions of hidden states**. Detailed results for OFA and GLEM are shown in `Table 2` and `Table 3` of the attached PDF, respectively. From the results, we observe that as the number of parameters increases, whether due to an increase in layers or an increase in the dimensions, there is no significant trend of improvement in model performance. This observation is more evident in the fitting of `Figure 1` of the attached PDF.
> >
> > - **Different scales of LLMs**: We use fixed GNNs and explore the scaling laws of LLMs in GraphLLM methods **using PLMs ranging from 4M to 13B**. The results are shown in `Table 4` of the attached PDF. For fairness, we introduce several versions of the same LLM-series for comparison, such as BERT-base and BERT-large. We find that increasing the scale of LLMs has less noticeable impact on model performance. Although OFA with LLaMA2-13B achieves the best results, the performance does not increase steadily with the increase in parameters, e.g., OFA with RoBERTa-large only achieves 58.3% ACC.
> >
> > In summary, it is hard to identify clear scaling laws, whether related to the scaling of GNNs or LLMs, which is consistent with Insight 4 in GLBench.
> >
> > *References*:
> >
> > [1] Neural Scaling Laws on Graphs. *Arxiv'24*

---

### Author Response · Authors · 2024-08-27
**General Response to All Reviewers**

Dear Reviewers,

We sincerely thank all the reviewers (eYMz, n5o4, 1oX9, rfpz) for their valuable feedback. We are glad that the reviewers appreciated the significance of our problem. i.e., GraphLLM (eYMz, n5o4, rfpz), the comprehensiveness of our experiments (eYMz, 1oX9, rfpz), the newly discussed zero-shot scenarios (eYMz, rfpz), some interesting insights such as scaling laws (n5o4, 1oX9), and the well-organized of our repository (eYMz, n5o4).

We have made every effort to faithfully address the comments in the responses. Specifically,

* We add the explanation for applying GraphLLM methods to non-text-attributed graphs. (eYMz)
* We add more rigorous and accurate experiments to analyze the scaling laws shown in GraphLLM from both GNN and LLM perspectives. (eYMz, rfpz)
* We add more analysis and insights, specifically how transferability varies with different pre-training datasets and settings. (eYMz, rfpz)
* We add more descriptions of what is lacking in the evaluation of recent papers. (n5o4)
* We add three commonly used GraphSSL methods in our comparison with GraphLLM methods. (1oX9)
* We add the explanation of how we use GraphSSL methods for zero-shot scenarios. (1oX9)
* We add findings and insights into why our "training-free" method can even outperform GraphLLM methods tailored for zero-shot scenarios that require pre-training. (1oX9)

We are willing to incorporate the suggested modifications in the next manuscript version. As the deadline for discussion is fast approaching, we would greatly appreciate it if you could allocate some time to review our responses.

Thanks for all the reviewers' time again.

Best regards,

Authors

---

### Decision · Program_Chairs · 2024-09-26

**Decision:**

Accept (Poster)

**Comment:**

This paper proposes GLBench, which is a comprehensive benchmark for evaluation of graph based LLM in both supervised and zero-shot learning settings. It further conducts extensive experiments on three groups of models which include GNNs, Pretrained Language Models, graph based LLMs (GraphLLMs) offering valuable insights and findings. These would serve as a guide to future research.

All the reviewers value the contribution of this work including a robust framework/benchmark to evaluate GraphLLMs models and useful findings inspiring the future research based on extensive empirical experiments. Concerns raised by reviewers were addressed during discussions.

I’d recommend its acceptance.